# Active twisting for adaptive droplet collection

Yifan Yang[1,2], Zhijun Dai[1,2], Yuzhen Chen[1,2] & Fan Xu [1] ✉

Many xeric plant leaves exhibit bending and twisting morphology, which may contribute to their important biological and physical functions adapted to drought and desert conditions. Revealing the relationships between various morphologies and functionalities can inspire device designs for meeting increasingly stringent environmental requirements. Here, demonstrated on the biomimetic bilayer ribbons made of liquid crystal elastomers, we reveal that the stimulus-induced morphological evolution of bending, spiraling, twisting and various coupling states among them can be selectively achieved and precisely tuned by designing the director orientations in liquid crystal elastomer bilayers. The mathematical models and analytical solutions are developed to quantify the morphology selection and phase transition of these liquid crystal elastomer ribbons for material design, as confirmed by experiments. Moreover, we show that, under activation and control of external stimuli, the twisting configuration can be harnessed to effectively collect and guide the transportation of droplets, and enhance the structural stiffness for resisting wind blow and rainfall to achieve the optimal configuration for water collection. Our results reveal the interesting functions correlated with bending, spiraling and twisting morphologies widely present in the natural world, by providing fundamental insights into their shape transformation and controlling factors. This work also demonstrates a potential application with integrating morphogenesis–environment interactions into devices or equipments.

The absence of freshwater is becoming a severe issue worldwide, particularly in arid regions[1–4]. The microstructures of some xeric plants such as cactus[5,6], nepenthes[7,8] and *Burkheya purpurea*[9] allow them to effectively collect droplets from fog. These plants provide inspiration and guidance for designing eco-friendly water-harvesting devices and addressing the challenges of drought climate[10–12]. Prior works have focused mainly on surface microstructural properties of leaves such as hydrophobicity[7,13], while macroscale leaf morphology of xeric plants may also play an important role for adapting to harsh environmental conditions. Observations suggest that many xeric plant leaves

are ribbon-like slim and exhibit helicoid-like twisting (for example, seedpod[14,15]) or bending–twisting coupled spiral shape (for example, *Erodium* awns[16,17] and sand lily[18]). The morphology of ribbon-like leaves is often dominated by the intrinsic curvature inducing bending or twisting, microscopically generated by growth-induced internal stress and strain mismatch between different textural layers of tissues[19–21]. Essentially, microscopic sources of these misfits must be anisotropic, and their principal directions are not consistent with the geometric characteristic directions (for example, length) of the ribbon-type structures[22,23]. Unraveling the underlying mechanism is key to precisely

[1]Institute of Mechanics and Computational Engineering, Department of Aeronautics and Astronautics & College of Intelligent Robotics and Advanced Manufacturing, Fudan University, Shanghai, People's Republic of China. [2]These authors contributed equally: Yifan Yang, Zhijun Dai, Yuzhen Chen. ✉e-mail: fanxu@fudan.edu.cn

**Fig. 1 | Various morphologies of ribbon-like geometry. a**, The bending shape of *Albuca namaquensis*. The spiral shape of *Ornithogalum concordianum*. The twisting shape of *Persoonia helix*. The cross-section of *Persoonia helix* shows the layered structure and anisotropy property. **b**, DIW of a LCE bilayer. The upper and lower layers can be programmed with different alignment orientations. **c**, Biomimetic bending ($\theta_u = 0°$, $\theta_l = 90°$), spiral ($\theta_u = 120°$, $\theta_l = 30°$) and twisting shapes ($\theta_u = 135°$, $\theta_l = 45°$) can be flexibly achieved by designing the orientations of LCE director between both layers during the DIW process. The blue- and yellow-colored boxes in the insets correspond to the director orientations of the upper and lower layers shown in **b**, respectively. **d**, A bionic active LCE cluster that can change its twisting morphology adapted to environmental cues, for example, the active twisting of strips adaptively increases upon heating to enhance droplet collection capability.

predicting their morphogenesis and further assisting in the design of biomimetic devices[24]. However, due to complex constitution and highly nonlinear deformation, theoretical understanding and prediction of their intriguing morphology formation and phase transition among twisting, bending and spiral shapes remains a challenge. Here, we reveal, both experimentally and theoretically, the morphogenesis pathways of bending, spiraling and twisting of ribbon-like bilayer structures and provide a phase diagram for morphing pattern selection. We demonstrate that the morphing interacted with and was tuned by environmental conditions, which can enable interesting functions such as effective water collection. Inspired by the layered structure and anisotropy properties of *Persoonia helix* (Fig. 1), we use liquid crystal elastomer (LCE)[25–28] bilayers with environmental stimuli (such as heating, lighting and raining) to mimic active anisotropic spontaneous strains between different layers upon growth. Combined with differential geometry and continuum theory based on a non-Euclidean shell model, we derive analytical solutions of distinguished deformed shapes (twisting, bending and spiral) and draw a complete closed-form phase diagram on morphing pattern selection by director orientations in different anisotropic swelling sublayers, confirmed by our experiments. We further provide an analytical solution of bending stiffness of twisting ribbons and experimentally show that the twisting configuration can dramatically enhance the structural bending stiffness, allowing xeric plants to resist severe windy environments. Guided by these insights, we design a bionic ribbon-type cluster that can mimic *Persoonia helix* to collect droplets by effectively transporting the water from leaves to roots along the twisted curvature, which can respond to environmental cues such as heat, light and humidity to adaptively transform its twisting morphology and, hence, water collection capability.

## Results

### Biomimetic LCE bilayer ribbons

Observations on the cross-section of xeric plant leaves in Fig. 1a suggest that the leaves mainly consist of an upper and a lower epidermis with a hydrogel-like soft core connection. Inspired by this, we apply

direct ink writing (DIW) technology[29] to fabricate the bilayer structures with designed arrangements (Fig. 1b). During the printing process, the orientation of the LCE director aligns with the LCE printing direction. By carefully designing the director orientation of the upper and lower layers through adjusting the three-dimensional (3D) printing direction, external stimuli such as heat, light and rain, can drive the strips to exhibit classic morphologies of xeric plant leaf, such as bending, spiral and twisting modes (Fig. 1c), due to the mismatch of spontaneous anisotropic deformations between bilayers. Then, we assemble several LCE strips into a biomimetic xeric plant (Fig. 1d), which allows adaptive deformations in response to environmental stimuli and enables functions such as water collection and wind resistance.

### Morphing mechanism and phase diagram for shape selection

We develop a bilayer model to understand the underlying morphing mechanism and to effectively predict the morphogenesis process. A ribbon-like elastic structure is commonly characterized as an intermediate object between rod (one-dimensional) and sheet (two-dimensional, 2D), which exhibits remarkable flexibility, sensitively exciting large-amplitude motion and deflection induced by small strains[30,31]. We consider a shell of thickness $H$ consisting of two sublayers with equal thickness $h = H/2$ (Supplementary Fig. 1e). Any material point of the bilayer is described by curvilinear coordinates $\mathbf{x} = (x_1, x_2, x_3)$, and the bilayer stays perfectly bonded at the interfaces. The material position in the initial undeformed configuration is denoted as $\bar{\mathbf{r}}(\mathbf{x})$, while $\mathbf{r}(\mathbf{x})$ represents the current state upon deformation. Considering the Kirchhoff–Love assumption, the material point $\mathbf{r}(\mathbf{x})$ can be written in terms of a normal offset from the mid-surface, $\mathbf{r}(x_1, x_2, x_3) = \mathbf{R}(x_1, x_2) + x_3\mathbf{N}(x_1, x_2)$, where $\mathbf{R}(x_1, x_2) = \mathbf{r}(x_1, x_2, 0)$ and $\mathbf{N}$ is the unit normal of the mid-surface. The bilayer system can be viewed as two flat sheets glued together, sharing the mid-surface ($x_3 = 0$). The metric tensor of the mid-surface[32] is defined as $g_{\alpha\beta} = a_{\alpha\beta} - 2x_3 b_{\alpha\beta} + \mathcal{O}(x_3^2)$, in which $a_{\alpha\beta}$ and $b_{\alpha\beta}$ are respectively the covariant components of the first and second fundamental forms evaluated at the mid-surface, expressed as $a_{\alpha\beta} = \mathbf{R}_{,\alpha} \cdot \mathbf{R}_{,\beta}$ and $b_{\alpha\beta} = \mathbf{R}_{,\alpha\beta} \cdot \mathbf{N}$, where the comma stands for differentiation with suffix coordinate. The 2D Green–Saint Venant strain tensor

of the shell is defined as $\varepsilon_{\alpha\beta} = (g_{\alpha\beta} - \bar{g}_{\alpha\beta})/2$. Within the Saint Venant–Kirchhoff material framework with Young's modulus $E$ and Poisson's ratio $\nu$, we introduce the 2D Saint Venant–Kirchhoff material norm as $\|\mathbf{A}\|_e^2 := \frac{E\nu}{1-\nu^2}\text{tr}^2(\mathbf{A}) + \frac{E}{1+\nu}\text{tr}(\mathbf{A}^2)$. The lower layer lies in the domain $x_3 \in [-H/2, 0]$ with the reference metric $\bar{\mathbf{g}}_l = \bar{\mathbf{a}}_l$, while the upper layer locates at $x_3 \in [0, H/2]$ with the reference metric $\bar{\mathbf{g}}_u = \bar{\mathbf{a}}_u$. Therefore, the total strain energy can be divided into two parts,

$$\mathcal{P} = \frac{1}{2}\int_S \int_{-H/2}^0 \|\bar{\mathbf{a}}_l^{-1}\boldsymbol{\varepsilon}_l\|_e^2 dx_3 d\bar{S} + \frac{1}{2}\int_S \int_0^{H/2} \|\bar{\mathbf{a}}_u^{-1}\boldsymbol{\varepsilon}_u\|_e^2 dx_3 d\bar{S}. \quad (1)$$

Selecting the appropriate combination of $\bar{\mathbf{a}}_l$ and $\bar{\mathbf{a}}_u$, the bilayer system (1) can be viewed as a curved monolayer with mid-surface reference fundamental forms $\mathbf{a}_0$ and $\mathbf{b}_0$. An intuitive choice would be $\mathbf{a}_0 = (\bar{\mathbf{a}}_l + \bar{\mathbf{a}}_u)/2$ and $\mathbf{b}_0 = (\bar{\mathbf{a}}_l - \bar{\mathbf{a}}_u)/2H$ (ref. 14). However, the exact solution based on energy equivalence reads[33]

$$\begin{aligned} \mathbf{a}_0 &= \frac{1}{2}(\bar{\mathbf{a}}_l + \bar{\mathbf{a}}_u), \\ \mathbf{b}_0 &= \frac{3}{4H}(\bar{\mathbf{a}}_l - \bar{\mathbf{a}}_u). \end{aligned} \quad (2)$$

Substituting equation (2) into equation (1), the strain energy can be simplified as

$$\mathcal{P} \propto \frac{1}{2}\int \left[\frac{H}{4}\|\mathbf{a}_0^{-1}\Delta\mathbf{a}\|_e^2 + \frac{H^3}{12}\|\mathbf{a}_0^{-1}\Delta\mathbf{b}\|_e^2\right] d\bar{S}, \quad (3)$$

in which $\Delta\mathbf{a} = \mathbf{a} - \mathbf{a}_0$ and $\Delta\mathbf{b} = \mathbf{b} - \mathbf{b}_0$ denote the change of metric tensors upon deformation. For simplicity, we assume that the Young's modulus and Poisson's ratio of both layers keep consistent, while the LC orientation can vary between both layers, but uniformly distributed in each LCE layer. The LC director is aligned in the $x_1 O x_2$ plane (planar alignment, that is, $\mathbf{n} = (\cos\theta, \sin\theta, 0)^T$), as shown in Fig. 1b. When subject to thermal load, the order degree of LC molecules decreases, leading to the spontaneous contraction along and expansion perpendicular to the LC director[34,35]. The spontaneous thermal expansion coefficients along the principal directions are denoted by $\alpha_\perp$ and $\alpha_\parallel$, respectively. We define the extension ratios in the direction parallel and perpendicular to the director as $\lambda_\parallel = 1 + \alpha_\parallel\Delta T$ and $\lambda_\perp = 1 + \alpha_\perp\Delta T$, where $\Delta T = T - T_0$ is the thermal change with respect to the reference temperature $T_0$. Hence, the spontaneous expansion can be regarded as the 'rest metrics' of each layer in the reference configuration, given by

$$\bar{\mathbf{a}}_i = \mathbf{R}(\theta_i)\begin{bmatrix} \lambda_\parallel^2 & 0 \\ 0 & \lambda_\perp^2 \end{bmatrix}\mathbf{R}^T(\theta_i), \quad (4)$$

where $i = l, u$ denotes the position of sublayer and $\mathbf{R}(\theta_i)$ is the 2D rotation matrix. Substituting equation (4) into equation (2) yields the equivalent fundamental forms $\mathbf{a}_0$ and $\mathbf{b}_0$ in the reference configuration

$$\begin{aligned} \mathbf{a}_0 &= \cos^2\frac{\Delta\theta}{2}\mathbf{R}(\hat{\theta})\begin{bmatrix} \lambda_\parallel^2 & 0 \\ 0 & \lambda_\perp^2 \end{bmatrix}\mathbf{R}(\hat{\theta})^T + \sin^2\frac{\Delta\theta}{2}\mathbf{R}(\hat{\theta})\begin{bmatrix} \lambda_\perp^2 & 0 \\ 0 & \lambda_\parallel^2 \end{bmatrix}\mathbf{R}(\hat{\theta})^T, \\ \mathbf{b}_0 &= \frac{3(\lambda_\parallel^2 - \lambda_\perp^2)}{4H}\mathbf{R}(\hat{\theta})\begin{bmatrix} 0 & \sin\Delta\theta \\ \sin\Delta\theta & 0 \end{bmatrix}\mathbf{R}(\hat{\theta})^T, \end{aligned} \quad (5)$$

in which $\hat{\theta} = (\theta_u + \theta_l)/2$ is the average director angle and $\Delta\theta = \theta_u - \theta_l$ represents the director angle difference.

Due to the presence of rest metrics, the bilayer tends to deform into a stress-free state, which minimizes the strain energy (3) without external constraints. We first consider a simple case that the director of the upper layer is parallel to the $x_1$ axis, while that of the lower layer is along the $x_2$ axis, that is, $\theta_u = 0°$ and $\theta_l = 90°$. Because the extension ratio is close to 1, we have $\mathbf{a}_0 \approx \mathbf{I}$. The deformed configuration is a classical saddle surface. However, when $W/L \ll 1$ (as a strip), the deformation along the $x_2$ axis can be neglected, and thus the deformed curvature satisfies

$$\frac{1}{\rho} = \frac{3(\lambda_\perp^2 - \lambda_\parallel^2)}{4H}. \quad (6)$$

The solid line in Fig. 2a represents this theoretical prediction, in good agreement with experiments and finite-element simulations. It can be observed that the bending angle $\alpha$ of the strip linearly grows with the increase of thermal variation $\Delta T$. Moreover, equation (6) can be reduced to $1/\rho = 3(\alpha_\perp - \alpha_\parallel)\Delta T/2H$ by neglecting higher-order terms, which recovers the Timoshenko's beam theory[36].

We next consider an LCE bilayer whose director angles are supplementary, that is, $\theta_u + \theta_l = 180°$. The Gaussian curvature[37] reads $K_G = \det(\mathbf{b}_0)/\det(\mathbf{a}_0)$. By computing the parametric equation of twisting surface, the analytical solution of Gaussian curvature on the centerline can be expressed as $\bar{K}_G = -k^2$, where $k$ stands for the wave number (Supplementary Section I). Combining with equation (5), we have

$$K_G \approx \bar{K}_G \Rightarrow k \approx \frac{3\sin\Delta\theta(\alpha_\perp - \alpha_\parallel)\Delta T}{2H}. \quad (7)$$

When $\Delta\theta = 90°$, the wave number $k$ reaches its maximum. The total twisting angle along the strip length is defined as $\beta = 180kL/\pi = 270(\alpha_\perp - \alpha_\parallel)\Delta TL\sin\Delta\theta/\pi H$. Figure 2b indicates that the twisting angle $\beta$ nonlinearly varies with the director angle difference $\Delta\theta$, remarkably consistent with theoretical prediction in equation (7). Moreover, when $\Delta\theta = 90°$, for example, $\theta_u = 135°$ and $\theta_l = 45°$, the twisting angle $\beta$ reaches its maximum value. Figure 2c plots the twisting angle $\beta$ with respect to thermal change $\Delta T$.

We further explore the occurrence of bending–twisting coupling helical shape. Let $\kappa_1$ and $\kappa_2$ be the principal curvatures of the mid-surface, and thus the Gaussian curvature can be expressed as $K_G = \kappa_1\kappa_2$ and mean curvature reads $H_{avg} = (\kappa_1 + \kappa_2)/2$. With small extension assumption $\mathbf{a}_0 \approx \mathbf{I}$, we have $K_G \approx \det(\mathbf{b}_0)$ and $H_{avg} \approx \text{tr}(\mathbf{b}_0)/2$. Therefore, the principal curvatures $\kappa_1$ and $\kappa_2$ can be regarded as the eigenvalues of $\mathbf{b}_0$. By computing the eigenvalues and eigenvectors of $\mathbf{b}_0$, one obtains principal curvatures $\kappa_{1,2} = \pm b_0 = \pm 3(\lambda_\parallel^2 - \lambda_\perp^2)\sin\Delta\theta/4H$ Then, the angle $\Phi$ and radius $R$ of bending–twisting helical deformation can be determined (Supplementary Section I) via

$$\begin{aligned} \Phi &= \arctan\frac{2\sin\phi\cos\phi}{\cos^2\phi - \sin^2\phi} = 2\phi, \\ R &= \frac{\cos 2\phi}{b_0}, \end{aligned} \quad (8)$$

where the angle between the principal direction and coordinate system is denoted as $\phi = \hat{\theta} + \pi/4$. Note that the deformed helix angle depends only on the microscopic director orientation and is irrelevant to thermal change. When $\cos^2\phi - \sin^2\phi = 0$, the helix angle $\Phi$ vanishes based on equation (8). One case satisfying this condition is that the upper and lower directors are supplementary, that is, $\theta_u + \theta_l = 180°$, and thus the bilayer ribbon exhibits a pure twisting deformation. Figure 2d illustrates the relation between thermal change $\Delta T$ and helical radius $R$ in the bending–twisting coupling spiral mode ($\theta_u = 120°$, $\theta_l = 30°$). With the increase of thermal change, the helical radius of the bilayer decreases. Detailed morphology evolutions in Fig. 2 are provided in Supplementary Section III. Note that the difference between theory and experimental results at larger $\Delta T$ (Fig. 2c,d)

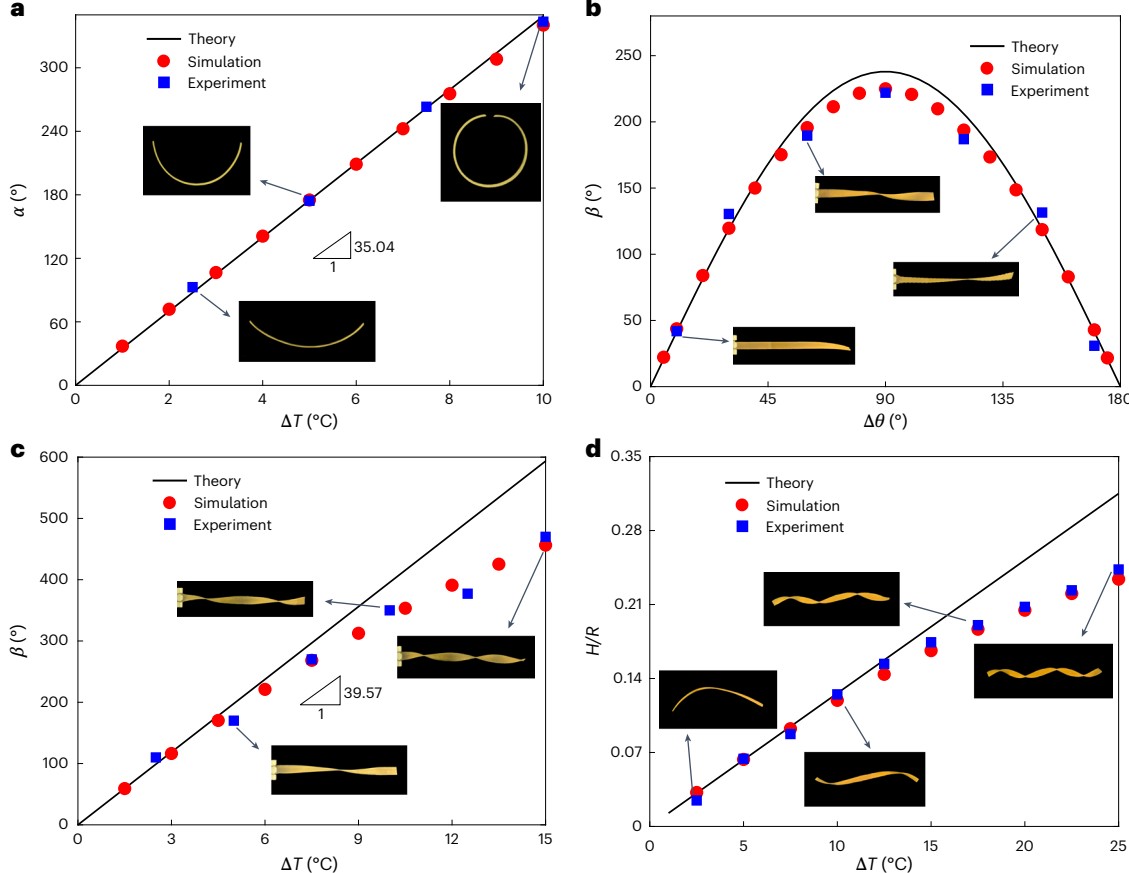

**Fig. 2 | Diverse deformed configurations of LCE bilayers. a**, The bending angle $\alpha$ of LCE bilayer strips as a function of thermal load $\Delta T$, where $\theta_u = 0°$, $\theta_l = 90°$. The solid line represents the analytical solution, while blue squares and red circles denote experiments and numerical results, respectively. **b,c**, The twisting angle $\beta$ of LCE bilayer strips varies with respect to the direction vector angle difference of LCE bilayer $\Delta\theta$ (**b**) and the thermal load $\Delta T$ (**c**). The temperature difference in **b** remains $\Delta T = 15$ °C, while the director orientation in **c** takes $\theta_u = 135°$, $\theta_l = 45°$. **d**, The spiral radius $R$ in the bending–twisting coupling mode as a function of thermal load $\Delta T$, where $\theta_u = 120°$, $\theta_l = 30°$.

can be attributed to the increasing geometric nonlinearity. As temperature arises, additional bending deformation along the width direction partially relieves the twisting or spiral deformation predicted by the theory.

Based on our theory that is validated both experimentally and numerically, the morphing shapes of LCE bilayer strips are determined primarily by the difference of director orientations between two sublayers and are independent of thermal change. The underlying mechanism of morphing configuration upon deformation is inherently related to the anisotropic mismatch strain generated by the director difference between bilayers. We draw a phase diagram in Fig. 3 to quantitatively reveal the determination of director orientations on pattern selection. When the director orientations of bilayers are identical, there is no out-of-plane deformation and the strip remains flat, corresponding to the dashed red line in Fig. 3. When the director of one layer is parallel to the $x_1$ axis and of the other layer is along the $x_2$ axis, the strip undergoes a pure bending deformation, denoted by pink squares. Moreover, when the director orientations of bilayers are supplementary, that is, $\theta_u + \theta_l = 180°$, a twisting shape emerges, represented by the solid blue line. The green region in the phase diagram corresponds to a bending–twisting coupling deformation. Note that our theory covers the prior predictions of bilayer deformation in refs. 14,38, where the authors concluded that the twisting deformation occurs only at $\theta_u = 45°$ and $\theta_l = 135°$, while our results suggest a much wider range of solution group satisfying $\theta_u + \theta_l = 180°$. The phase diagram can be used to quantitatively guide morphology designs of ribbon-like slender structures.

## Twisting cluster interacting with environment

We designed biomimetic active twisting clusters using LCE bilayers to demonstrate their environmentally responsive morphogenesis and explore potential functions. We fabricated a series of LCE biomimetic strips with different (twisting, bending or spiral) morphologies based on DIW technology, guided by the phase diagram in Fig. 3. Due to nozzle confinement of DIW, the alignment of LC molecules is parallel to the printing path, and thus LCE bilayers with different alignment orientations can be achieved by programming the nozzle moving direction. Carbon fibers or pliable iron wires were implanted into the LCE strips as their main stems. A 365-nm ultraviolet (UV) lamp was used for LCE curing. The detailed fabrication process can be found in Supplementary Section III. LCE plant clusters can interactively respond to environmental cues such as heat, rain and light, adaptively changing their shapes and droplet collection capability. As demonstrated in Fig. 4a), the plant cluster initially exhibits a relatively flat leaf structure at room temperature 28.1 °C. Upon heating to 101.4 °C, the 'leaves' actively morph into twisting shapes like *Peisoonia helix*, reaching an average twisting angle of 519.2° (Fig. 4b). Notably, the twisting degree is more pronounced near the heat sources, while the central region, situated at a larger distance from the heat sources, experiences less twisting deformation. Once the twist stabilized, rainfall was simulated. The raindrops absorb heat from the leaves, causing a decrease in leaf temperature to 86.9 °C and consequently inducing an untwisting deformation to an average twisting angle of 274.6° (Fig. 4c). Finally, we removed the heat source and allowed the sample to gradually cool down to room temperature, leading to further untwisting of the leaves back to the

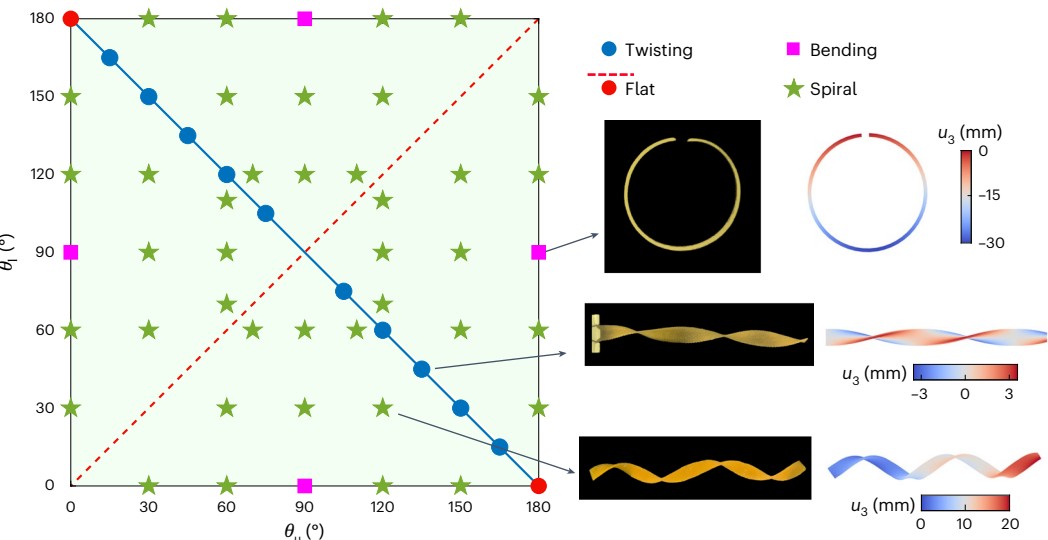

**Fig. 3 | Phase diagram on deformed morphologies of LCE bilayers.** The pink squares represent the pure bending mode, while the blue solid line stands for the pure twisting shape. The dashed red line denotes the flat state, while the remaining green region represents the bending–twisting coupled spiral mode.

initial configuration with an average residual twisting angle of 26.2° (Fig. 4d). In addition, we conducted a cyclic heating–raining–cooling process involving two instances of rainfall during continuous heating, simulating complex climate change in a harsh environment, as demonstrated in Supplementary Video 1. We further demonstrated that such active twisting response can be flexibly tuned by localized laser (450 nm, 3.07 W cm$^{-2}$) illumination (Fig. 4e–h). The leaf under laser illumination was twisted to 210.9° because of thermal effect in the irradiation area, while the others remained unchanged. When the laser was turned off, the twisted leaf reverted to the initial flat state (Fig. 4h). The intelligent environmental interaction of the biomimetic plant cluster is highlighted, which can enable promising applications based on tunable morphing.

We next investigated wind resistance of the twisting configuration. As shown in Fig. 4i, wind was applied to a twisting strip and a flat one, respectively. The maximum deflection angles under wind blow are 15° for the twisting leaf and 62° for the flat leaf. Moreover, after wind cessation, the twisting leaf can almost return to its initial vertical position, while a residual bending deformation persists in the flat leaf. This fact implies that the twisting configuration can dramatically increase the bending stiffness $\bar{k}$ of the strip and, thus, exhibits superior wind resistance, which can be quantitatively characterized by a dimensionless parameter

$$\bar{k} = \frac{96\beta^4 I_2}{\sqrt{36 C_2{}^2 (I_2 - I_3)^2 + C_3{}^2 I_3{}^2}}, \tag{9}$$

where $\beta$ denotes the twisting angle, $E$ is the Young's modulus, $I_2$ and $I_3$ represent the inertia moments of the strip, and $C_2$ and $C_3$ are coefficients related to the twisting angle (see Supplementary Section I for detailed derivations). As plotted in Fig. 4j, as the twisting angle increases, the bending stiffness substantially increases, which suggests that the twisting morphology of most xeric plants such as *Peisoonia helix* holds superior advantage in resisting wind blow in harsh environments.

## Twisting for adaptive water collection
We further explored the droplet collection and directional transport based on twisting morphology. As depicted in Fig. 5a, when a droplet lands on the surface of the twisting leaf, it can smoothly slide from the top to the bottom, resembling the descent on a twisting slide. Notably, both surfaces of the strip show identical droplet guiding characteristics,

ensuring its ability to collect and convey droplets from various directions. We next compared droplet collection efficiency between the twisting and bending shaped strips that were affixed to the apex of slender rods, positioned inside graduated cylinders. The cylinders were sealed at the top with a lid featuring a small aperture, thereby ensuring the collection of only those droplets flowing along the root from the leaf. To assess the omnidirectional water-collecting capabilities of the samples in various raining conditions, we uniformly rotated the samples using a rotating exhibition platform. Rainfall simulation was achieved through an irrigation apparatus. Comparative experiments demonstrated that the twisting leaf exhibits a much better water-collecting capability (6.3 ml for twisting morphology and 3.0 ml for bending shape in Fig. 5b). The twisting configuration can efficiently direct the collected droplets on the leaf surface toward root storage, while droplets falling on the bent leaf surface are bifurcated, with a portion being stored and the remainder sliding down along the leaf tip. We further looked into the collective effect on water collection by examining biomimetic permanently preset twisting clusters and bending shaped clusters. The comparisons between twisting and bending clusters in single-level (Fig. 5c) and multilevel (Fig. 5d) configurations indicate that the twisting clusters display a superior water-collecting performance (Supplementary Video 2).

Moreover, we performed droplet collection experiments in windy environment, aiming to scrutinize the water collection attributes of biomimetic plants amid extreme weather conditions in arid regions. As illustrated in Fig. 5e, flat leaves, owing to their intrinsic weak bending stiffness, swiftly succumbed to the wind, making it nearly impossible for them to transport raindrops toward the roots. By contrast, twisted leaves inherently exhibit superior resistance to bending deformation, maintaining a relatively upright posture even under severe wind blow, which ensures a notable efficiency in rainwater collection and transportation. To further demonstrate the tunable water collection capacity of the artificial plant interacting with environment change, we perform adaptive drop collection experiments. As shown in Fig. 5f, the cluster with untwisted leaves collects 2.4 ml of water under rain at room temperature (25.8 °C). Upon heating to 79.0 °C, leading to a spontaneous twisting of leaves, the amount of water collection increases to 7.3 ml over the same period. When the temperature reverts back to the initial state, the water collection capacity decreases adaptively (Supplementary Video 2). We further show in Fig. 5g the flexible tunability of morphing-affected droplet collection by localized laser illumination (1.65 W cm$^{-2}$, the upper part of the cluster). Under photo illumination,

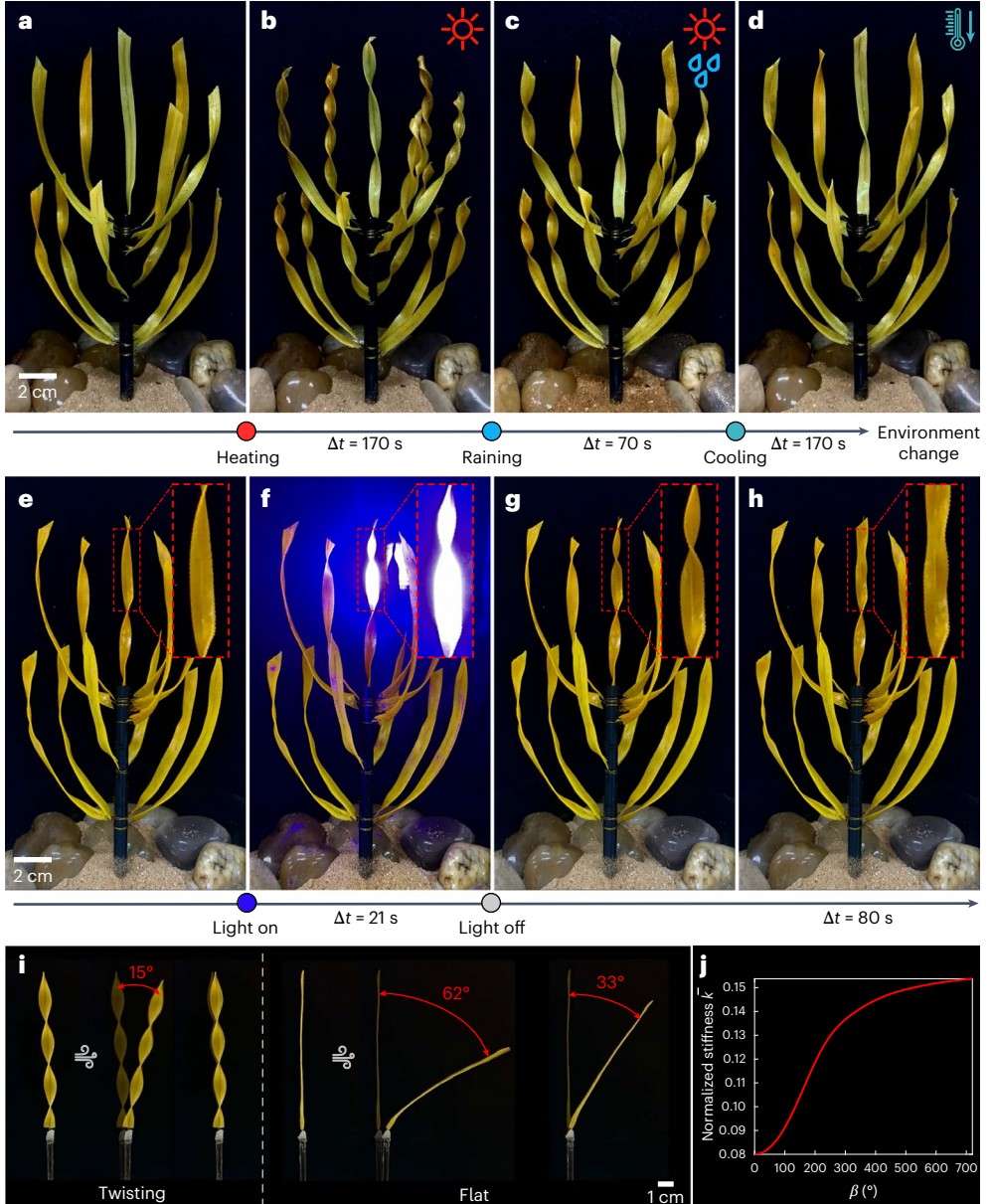

**Fig. 4 | Morphing active twisting plants. a–d,** A biomimetic LCE plant interacted with environmental cues: initial state at room temperature (**a**); active twisting of the 'leaves' upon heating (**b**); untwisting of the leaves after rainfall (**c**); complete untwisting upon cooling to room temperature (**d**). **e–h,** Flexible twisting morphology change of a single leaf under localized laser (450 nm) illumination:

initial state (**e**); active twisting upon laser illumination (**f**); immediate state upon turning off the laser (**g**); untwisted shape after recovery (**h**). **i,** Wind resistance performance of a twisted leaf and a flat one. **j,** Variation of the normalized bending stiffness $\bar{k}$ of the LCE strip with respect to the twisting angle $\beta$. With the rise of the twisting angle, the bending stiffness dramatically increases.

the ability of water collection increases dramatically (from 2.4 ml to 6.1 ml) due to thermally induced twisting deformation of active leaves. Upon light off, the leaves revert back to their original flat shape, leading to the reduction of water collection capacity (Supplementary Video 3). Overall, we revealed that the twisting morphology not only is efficient in droplet collection but also dramatically enhances the structural stiffness for sustaining harsh environmental condition.

## Discussion

We have revealed that the twisting configuration of most xeric plant leaves contributes to their effective droplet collection and wind resistance. More than natural xeric plants, the morphing of LCE ribbons can be selectively tuned among bending, twisting and bending–twisting spiral morphologies guided by the model we developed. Inspired by this morphing mechanism, we have designed an artificial cluster that can

realize tunable droplet collection in different environments, although some issues remain to be explored in future. The mechanical response when droplets impact the bilayers may need further investigation, as the dynamic interaction between falling droplets and the leaf surface would affect water collection efficiency. Optimizing bilayer stiffness and interface properties could potentially enhance droplet collection performance. In addition, our current theoretical model, although effective, could be extended to incorporate director gradient distribution and variable-dimension configurations that would enable more sophisticated and intelligent control strategies.

Material durability in extreme environments would be a key challenge for practical long-term applications. Future efforts could be made to explore combining LCE ribbons with protective and hydrophobic coatings to improve UV resistance and environmental durability, as well as enhancing droplet transport efficiency. Moreover,

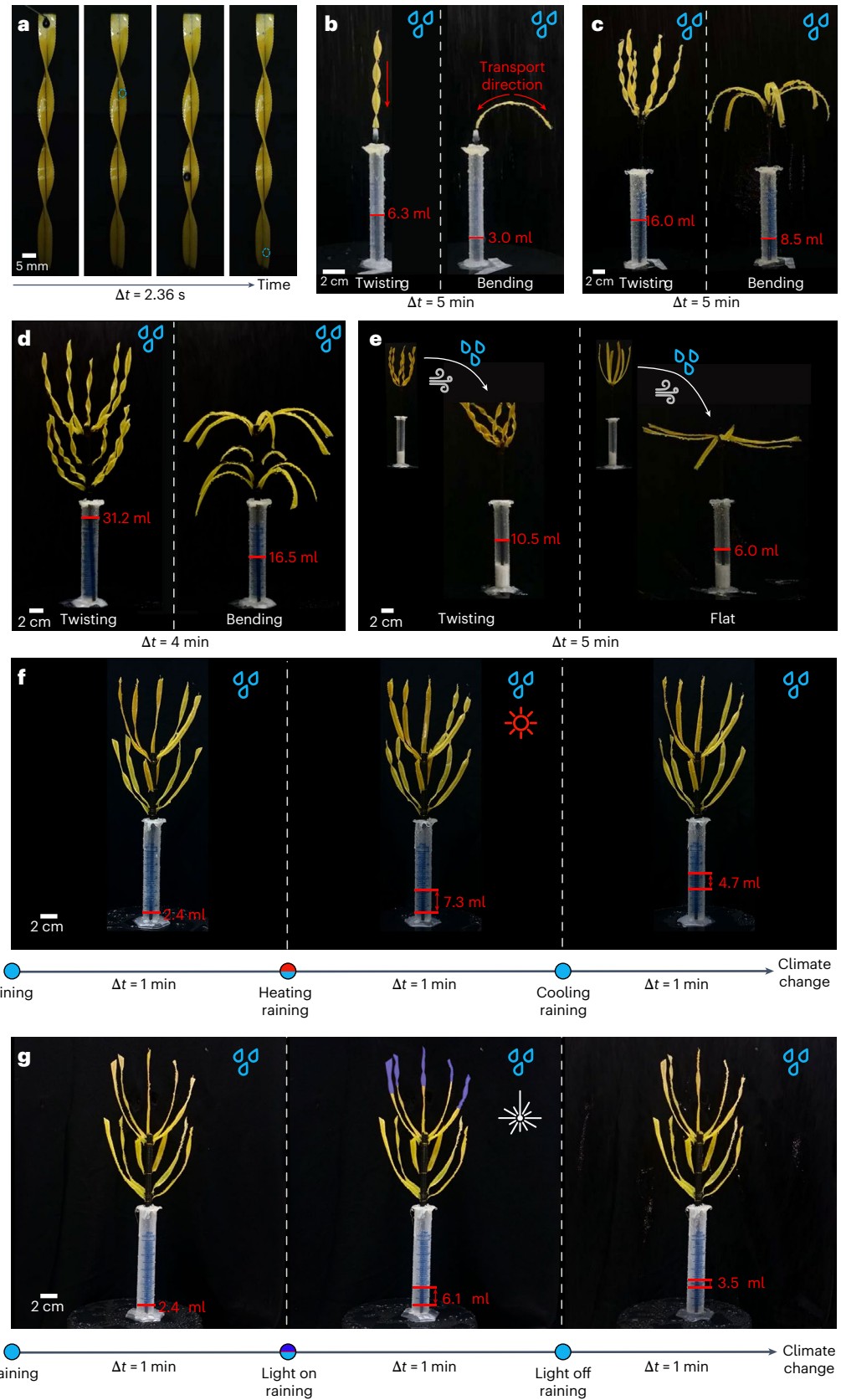

**Fig. 5 | Droplet collection experiments. a**, Directional transport of droplets on the surface of a twisted strip. **b**–**d**, Water collection of a single LCE 'leaf' (**b**), a cluster (**c**) and a bionic multilevel cluster (**d**), illustrating the superior water collection capacity of twisting morphology compared with bending configuration. **e**, Comparison of droplet collection between a twisted cluster and a bent one under windy environment. **f**, Environment-adaptive droplet collection of the intelligent cluster with active morphology changes. **g**, Light-tuned droplet collection of the intelligent cluster with local morphology changes.

the development of hybrid material systems integrating LCE with photovoltaic components could result in multifunctional adaptive surfaces capable of collecting both energy and water simultaneously. Such an integrated strategy would expand the practical implications of this research beyond water collection to comprehensive resource management in water-energy-constrained regions.

## Methods

### Fabrication of LCE bilayers

RM82 (1,4-bis-[4-(6-acryloyloxyhexyloxy)benzoyloxy]-2-methylbenzene; Sigma-Aldrich), *n*-butylamine (Sigma-Aldrich) and Irgacure I-369 (Sigma-Aldrich), were used in experiments for fabricating LCE bilayers[39]. We mixed liquid crystal monomer RM82 and *n*-butylamine in a 1.1:1 molar ratio, added 1.5 wt.% photoinitiator Irgacure I-369 and blended with heat until homogeneous. The mixture was vacuum-treated for 40 min, transferred to a printing syringe and oligomerized at 75 °C for 12 h. Note that the phase transition temperature of the LCEs is 105 °C, which is consistent with the similar synthesis methods[39]. The LCE ink was loaded into a DIW 3D printer (Adventure, 3D-LB-Printer-0050) with a 410-µm nozzle at 70 °C. Printing was performed at 0.3 MPa driving air pressure and 800 mm min⁻¹ speed, with G-code controlling the director orientation. Samples were UV-cured (ZLUVLAMI, 50 W, 365 nm) for 15 min on each side.

### Fabrication of active leaves

As illustrated in Supplementary Fig. 9, a bottom layer was initially deposited through 3D printing. Subsequently, a carbon fiber rod (diameter ≈0.5 mm) was positioned along the central axis on the bottom layer, and then printing an upper layer, covering the rod. The entire sample was then molded after curing under UV light, which was used for droplet collection experiments. Heating the sample can induce a twisting deformation. Moreover, after completing step C in the printing process (Supplementary Fig. 9), the sample underwent semicuring. While retaining a certain degree of fluidity, the carbon fiber rod can be extracted from the sample, forming a channel. Subsequently, the sample was fully cured, and a metal wire (diameter ≈0.3 mm) was inserted into the channel. The plasticity of the wire allows the bending of the sample into a curved shape. Due to the presence of a small gap between the channel walls and the wire, the LCE leaves can undergo reversible deformations without considerable constraints from the wire.

### Experimental setup

To characterize $\alpha_\parallel$ and $\alpha_\perp$ within the LCE monolayer, square LCE sheets (50 mm × 50 mm × 1.2 mm) were heated incrementally by 5 °C with 10-min dwell periods on a heating bed (LICHEN, LC-DB-1DA). Length and width changes were used to calculate thermal expansion coefficients $\alpha_\parallel = (W_t - W)/W\Delta T$ and $\alpha_\perp = (L_t - L)/L\Delta T$, where $W_t$ and $L_t$ denote the width and length at different temperatures (Supplementary Table 1).

The droplet collection setup (Supplementary Fig. 10) includes a rotating platform (7 rpm) and an irrigation water gun connected to a faucet. Simulated rainfall had an average intensity of 24.4 ml min⁻¹. Wind speed was set at 5 m s⁻¹ for wind resistance tests (Fig. 4i) and 7 m s⁻¹ for droplet collection (Fig. 5e) in windy conditions. Each experiment was repeated five times. Data in Supplementary Table 2 show that twisted leaves collect droplets more effectively than the leaves with pure bending configuration. Even in windy environments, the twisting shape dramatically (almost doubly) enhances leaf stiffness, which also helps droplet collection. For adaptive droplet collection experiments in Fig. 5f,g, each collection duration lasted 1 min.

### Numerical method

We performed finite-element simulations based on a solid-shell model[40,41] to explore the morphing of LCE bilayers. The eight-node brick element was used for discretization, which avoided introducing rotational degrees of freedom and, thus, does not require complex finite-rotation updates. The enhanced assumed strain and assumed natural strain methods were used to overcome locking effects. Previous studies have shown that this approach provides high accuracy and excellent performance for problems related to thin-walled soft structures. We applied the standard Newton–Raphson method to solve the nonlinear system. The mesh convergence was carefully examined, and mesh density was set as 50 × 4 × 2 for bending mode, 50 × 8 × 2 for twisting mode and 150 × 10 × 2 for spiral mode. In the calculations, the temperature difference $\Delta T$ was introduced as the loading, gradually increasing from 0 to the specified temperature difference. Details on numerical implementation can be found in Supplementary Section II.

## Data availability

Source data are provided with this paper.

## Code availability

The code used in this study is available via Zenodo at https://doi.org/10.5281/zenodo.14936583 (ref. 42).

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

## Acknowledgements

This work is supported by the National Natural Science Foundation of China (grant nos. 12425204, 12122204 and 12372096), Shanghai Pilot Program for Basic Research-Fudan University (grant no. 21TQ1400100-21TQ010), Shanghai Shuguang Program (grant no. 21SG05), Shanghai Municipal Education Commission (grant no. 24KXZNA14), China Postdoctoral Science Foundation (grant no. 2024M750510) and Shanghai Post-doctoral Excellence Program (grant no. 2023208).

## Author contributions

F.X. conceived the idea and designed the research. Y.Y., Z.D. and Y.C. conducted the experiments. Y.Y. and F.X. developed the theoretical models. Y.Y. performed the numerical computations. Y.Y., Z.D. and F.X. interpreted the results. Y.Y., Z.D. and F.X. wrote the manuscript. All the authors provided helpful discussions.

## Competing interests

The authors declare no competing interests.

## Additional information

**Correspondence and requests for materials** should be addressed to Fan Xu.

