## [Peer Review File · Nature Computational Science]

Active twisting for adaptive droplet collection

Corresponding Author: Professor Fan Xu

Version 0:

Decision Letter:

**** Please ensure you delete the link to your author homepage in this e-mail if you wish to forward it to your co-authors. ****

Dear Professor Xu,

Your manuscript "Active twisting for adaptive droplet collection" has now been seen by 2 referees, whose comments are appended below. You will see that while they find your work of interest, they have raised points that need to be addressed before we can make a decision on publication.

The referees' reports seem to be quite clear. Naturally, we will need you to address ***all*** of the points raised.

While we ask you to address all of the points raised, the following points need to be substantially worked on:

- Following our referees' suggestion, please provide the requested technical details of your model, simulation and experiments.
- Please provide more quantitative discussions about your results and experimental/simulation settings.

Please use the following link to submit your revised manuscript and a point-by-point response to the referees' comments (which should be in a separate document to any cover letter):

Link Redacted

**** This url links to your confidential homepage and associated information about manuscripts you may have submitted or be reviewing for us. If you wish to forward this e-mail to co-authors, please delete this link to your homepage first. ****

To aid in the review process, we would appreciate it if you could also provide a copy of your manuscript files that indicates your revisions by making use of Track Changes or similar mark-up tools. Please also ensure that all correspondence is marked with your Nature Computational Science reference number in the subject line.

In addition, please make sure to upload a Word Document or LaTeX version of your text, to assist us in the editorial stage.

If you have any issues when updating your Code Ocean capsule during the revision process, please email the Code Ocean support team Cc'ing me.

To improve transparency in authorship, we request that all authors identified as 'corresponding author' on published papers create and link their Open Researcher and Contributor Identifier (ORCID) with their account on the Manuscript Tracking System (MTS), prior to acceptance. ORCID helps the scientific community achieve unambiguous attribution of all scholarly contributions. You can create and link your ORCID from the home page of the MTS by clicking on 'Modify my Springer Nature account'. For more information please visit www.springernature.com/orcid.

We hope to receive your revised paper within three weeks. If you cannot send it within this time, please let us know.

Best regards,

Jie Pan, Ph.D.
Senior Editor
Nature Computational Science

Reviewers comments:

Reviewer #1 (Remarks to the Author):

see attached.

Reviewer #2 (Remarks to the Author):

This paper presents the design of bio-inspired leaves using the bilayer liquid crystal elastomers. The authors present numerical models and experiments to support the idea. They also present a series of examples of how the deformation mode can be tuned by designing the orientations of the director in LCE bilayers. I read this paper with interest in the topic; overall, I found it to be a nice work combining computational efforts and experiments. However, the author should address the questions below before publication to make the paper easier to read and more informative. My main comment to the authors is to provide quantitative values of various parameters in experiments such as temperature, light intensity, and actuation/relaxation times, etc.

Detailed comments:

In Fig. 1c, the nematic alignments for twisting and spiral should be different, but the figure does not clearly show the difference. Please modify the figure or add a note in the caption about how the alignments of bilayers in these two cases are different.

Below Eq. (7): "Figure 2(b) indicates that the twisting angle β nonlinearly varies with the director angle difference $\Delta\theta$, remarkably consistent with theoretical prediction in Eq. (7)."

The relation of β and $\Delta\theta$ is not clearly given in Eq. (7). Please clarify this.

Fig. 2(c-d): Please add a clear note as to why the simulation and experiments deviate from theory at larger temperatures. Is there any relation between the deviation point and the nematic alignments in LCE layers? I recommend more analysis and comments on the physics of these observations.

The results throughout the paper are mainly descriptive referring to the experimental figures. I recommend authors add more quantitative values to support their descriptions. For example "Once the twist stabilizes, rainfall was simulated. The raindrops absorb heat from the "leaves", causing a decrease in leaf temperature and consequently inducing an untwisting deformation": Please provide quantities on how much the twist and untwist are. And what are the actuation and relaxation times?

Also, "Finally, we removed the heat source and allow the sample to gradually cool down to room temperature, leading to further untwisting of the "leaves" back to the initial configuration": what is the time scale of twisting subject to illumination and untwisting after removing illumination? What is the light intensity and wavelength?

Please also provide information about the phase transition temperature of the synthesized LCEs.

Please add the time scales in Fig. 4(a-h).

The authors should make comments on why the initial twisted configuration of leaves in Fig. 5d and 5f are different. If it depends on temperature, the authors should mention the temperature.

The authors should identify the temperatures or temperature changes in all experiments, particularly, in Fig. 4 and 5 (and any other figures as needed).

The authors should identify the light intensity and wavelength used in experiments in all Figures.

Please provide more details on mesh convergence in the Method section. What is the size of the converged mesh used in the simulation?

The authors should carefully review the equations, figures, and citations in SI for any potential typos. For instance, in SI, part C: "The general spatial orientation of LC director is denoted by a unit vector $\mathbf{n} = (\sin\phi\cos\theta, \sin\phi\sin\theta, \cos\phi)\mathbf{T}$, as shown in Fig. S1(b), where $\phi(u)$ denotes the angle between director and x3-axis, $\theta(u)$ is the angle between the projection of director on

x1Ox2 plane and x1-axis." Fig. S1(b) shows something different. Please also identify angles ϕ and θ in the figure.

Version 1:

Decision Letter:

Our ref: NATCOMPUTSCI-24-2296A

12th February 2025

Dear Dr. Xu,

Thank you for submitting your revised manuscript "Active twisting for adaptive droplet collection" (NATCOMPUTSCI-24-2296A). It has now been seen by the original referees and their comments are below. The reviewers find that the paper has improved in revision, and therefore we'll be happy in principle to publish it in Nature Computational Science, pending revisions to satisfy the referees' final requests on the code availability and to comply with our editorial and formatting guidelines.

We are now performing detailed checks on your paper and will send you a checklist detailing our editorial and formatting requirements in about two weeks. Please do not upload the final materials and make any revisions until you receive this additional information from us.

TRANSPARENT PEER REVIEW

Nature Computational Science offers a transparent peer review option for original research manuscripts. We encourage increased transparency in peer review by publishing the reviewer comments, author rebuttal letters and editorial decision letters if the authors agree. Such peer review material is made available as a supplementary peer review file. **Please remember to choose, using the manuscript system, whether or not you want to participate in transparent peer review.**

Thank you again for your interest in Nature Computational Science. Please do not hesitate to contact me if you have any questions.

Sincerely,

Jie Pan, Ph.D.
Senior Editor
Nature Computational Science

ORCID

Reviewer #1 (Remarks to the Author):

The authors have satisfactorily addressed all of my concerns. I am happy to recommend publication.

Reviewer #2 (Remarks to the Author):

Thanks to the authors for addressing the questions and revising the manuscript.
Please see a comment regarding the source code availability.

Reviewer #2 (Remarks on code availability):

I reviewed the codes that authors made available. I could not find any source code for the finite element simulations used in the paper. If the authors used any existing software, they should mention it; if they developed their own code, it would be beneficial to make it available for reproducing the data, even for a simple case presented in the manuscript. If they cannot share their finite element code, the claim that "The code used in this study is available with this paper" would be inaccurate, and they should instead provide all simulation details such as geometrical and material properties used in their simulations.

Version 2:

Decision Letter:

Dear Professor Xu,

We are pleased to inform you that your Article "Active twisting for adaptive droplet collection" has now been accepted for publication in Nature Computational Science.

Once your manuscript is typeset, you will receive an email with a link to choose the appropriate publishing options for your paper and our Author Services team will be in touch regarding any additional information that may be required.

Acceptance of your manuscript is conditional on all authors' agreement with our publication policies (see <https://www.nature.com/natcomputsci/for-authors>). In particular your manuscript must not be published elsewhere and there must be no announcement of the work to any media outlet until the publication date (the day on which it is uploaded onto our web site).

Before your manuscript is typeset, we will edit the text to ensure it is intelligible to our wide readership and conforms to house style. We look particularly carefully at the titles of all papers to ensure that they are relatively brief and understandable.

Once your manuscript is typeset, you will receive a link to your electronic proof via email with a request to make any corrections within 48 hours. If, when you receive your proof, you cannot meet this deadline, please inform us at rjsproduction@springernature.com immediately.

If you have queries at any point during the production process then please contact the production team at rjsproduction@springernature.com.

We welcome the submission of potential cover material (including a short caption of around 40 words) related to your manuscript; suggestions should be sent to Nature Computational Science as electronic files (the image should be 300 dpi at 210 x 297 mm in either TIFF or JPEG format). We also welcome suggestions for the Hero Image, which appears at the top of our [home page](http://www.nature.com/natcomputsci); these should be 72 dpi at 1400 x 400 pixels in JPEG format. Please note that such pictures should be selected more for their aesthetic appeal than for their scientific content, and that colour images work better than black and white or grayscale images. Please do not try to design a cover with the Nature Computational Science logo etc., and please do not submit composites of images related to your work. I am sure you will understand that we cannot make any promise as to whether any of your suggestions might be selected for the cover of the journal.

Best regards,

Jie Pan, Ph.D.
Senior Editor
Nature Computational Science

P.S. Click on the following link if you would like to recommend Nature Computational Science to your librarian: https://www.springernature.com/gp/librarians/recommend-to-your-library

** Visit the Springer Nature Editorial and Publishing website at www.springernature.com/editorial-and-publishing-jobs for more information about our career opportunities. If you have any questions please click here. **

Response to Reviewers

Title	Active twisting for adaptive droplet collection
Authors	Yifan Yang, Zhijun Dai, Yuzhen Chen, and Fan Xu
Journal	Nature Computational Science
Manuscript No.	NATCOMPUTSCI-24-2296

Reviewer #1

- **Comment (0)**

In this paper, the authors present a combined theoretical, numerical, and experimental approach to analyse a system of actively twisting morphing ribbons. The experimental setup that the authors devise and on which their model is based consists of two families of Liquid Crystal Elastomers, which actively contract upon heating and thus generate a change in shape of the thin structure. The authors first analyse how the angle between the two families of LCE impacts the resulting morphology upon heating, showing how different modes of deformation – bending, twisting, and spiraling – may be consistently achieved by varying the fibre angles. The authors then consider a potential application for this morphing system, the collection of water droplets. The authors show how tuning the morphology of the LCE ribbons provides a potential for smart collection of water drops, in which induced twist enables droplets to better be guided into a central point. It is also shown how induced twisting increases bending rigidity, thus enabling the structure to better withstand strong winds.

This paper is very interesting. The combined analysis is thorough, well presented, and convincing. The application the authors consider is unusual but intriguing. I have no idea how practical such a device would be, but I find it to be a very promising avenue of investigation. The work is worthy of publication. I have only some minor comments for the authors to consider.

- ✓ **Response (0)**

We thank the reviewer for the insightful, encouraging comments and strong support of the work. We appreciate the reviewer for his/her very careful reading and kind suggestions to help us improve the manuscript. All the concerns raised by the reviewer are addressed point-by-point in the following.

- **Comment (1)**

The agreement with theory, simulation, and experiment shown in Fig 2 is impressive. But the authors do not seem to comment on the discrepancy between theory and experiment in Fig 2 (c) and (d) at larger ΔT . Is this a consequence of nonlinearities? Some comment here would be appropriate.

- ✓ **Response (1)**

We thank the reviewer for the insightful comment. In the theoretical analysis, we obtained the relation between the Gaussian curvature of the strip centerline and spontaneous strain to characterize the strip

morphology evolution for both twist and spiral modes. However, at higher temperatures with larger deformations, the increasing geometric nonlinearity may induce bending along the width (W) direction, reducing the overall twisting or spiral deformation. This effect could explain why the theoretical results are slightly higher than the computational and experimental results in Fig. 2(c) and (d). We have added some discussions in the main text.

“Note that the difference between theory and experimental results at larger ΔT (Fig. 2(c)-(d)) can be attributed to the increasing geometric nonlinearity. As temperature arises, additional bending deformation along the width direction partially relieves the twisting or spiral deformation predicted by the theory.”

- **Comment (2)**

It would be useful to clearly define, in a half sentence say, the twisting angle. For those readers unfamiliar with the theory of ribbons, this term might not be clear, and the images that accompany Fig 2(b) do not really help to make clear what exactly is meant by β .

- ✓ **Response (2)**

This is a good point and we fully agree on this comment. We have provided the definition of β in the main text, which gives the explicit relation between the twisting angle and director angle difference.

“The total twisting angle along the strip length is defined as $\beta = 180kL/\pi = 270(\alpha_{\perp} - \alpha_{\parallel})\Delta TL \sin\Delta\theta/\pi H$.”

- **Comment (3)**

The authors show very clear the benefit of a twisted morphology both in collecting droplets and in resisting bending under a strong wind. What was less clear to me was this: is there a benefit to untwisting? In other words, if I wanted to design an optimal rain collector, clearly I should use twisted ribbons; but why do I care if they also untwist? Shouldn't I just use a permanently twisted material? I know that tunable morphology is an interesting feature on its own, but it wasn't clear to me, in this specific application, that there was a benefit to having a changing morphology. For instance, the authors compare the water collecting capabilities of a flat geometry against their morphing structure. But what if there were a third geometry that was rigidly twisted? Wouldn't such a structure perform better (collect more water) than the morphing structure? Note: this is not to negate the value of the study. But perhaps some discussion along these lines would help in clarifying the potential value.

- ✓ **Response (3)**

Thanks for this insightful comment. Indeed, twisted ribbons show superior wind resistance and water collection efficiency. The adaptive morphology serves a more general purpose: the twisting angle can be increased in dry environment to enhance water collection, while the twisting angle can be decreased

to prevent excessive water accumulation in wet conditions. Such environmental adaptability provides flexible active optimization that a permanently twisted structure cannot achieve. Future work will be focused on integrating such adaptive mechanism into specific practical applications.

We have revised the Conclusion section to highlight these advantages.

“Inspired by the morphing mechanism of xeric plants such as *Persoonia helix*, we have designed an artificial cluster that can realize tunable droplet collection in different environments. Unlike fixed twisted shapes, our active system enables dynamic morphing and stiffness tuning that interactively respond to environmental cues such as heat, rain, light and wind. These adaptive shape changes could allow active optimization by twisting to improve water collection in dry conditions while untwisting to prevent excess accumulation in wet environments, shedding light on the potential applications in adaptive water collection and directional transportation.”

Reviewer #2

- **Comment (0)**

This paper presents the design of bio-inspired leaves using the bilayer liquid crystal elastomers. The authors present numerical models and experiments to support the idea. They also present a series of examples of how the deformation mode can be tuned by designing the orientations of the director in LCE bilayers. I read this paper with interest in the topic; overall, I found it to be a nice work combining computational efforts and experiments. However, the author should address the questions below before publication to make the paper easier to read and more informative. My main comment to the authors is to provide quantitative values of various parameters in experiments such as temperature, light intensity, and actuation/relaxation times, etc.

- ✓ **Response (0)**

We thank the reviewer for the insightful, encouraging comments and very comprehensive summary of the work, by pointing out that “a nice work combining computational efforts and experiments”. We also thank the reviewer for his/her very careful reading and kind suggestions to help us improve the manuscript. All the concerns raised by the reviewer are addressed point-by-point in the following.

- **Comment (1)**

In Fig. 1c, the nematic alignments for twisting and spiral should be different, but the figure does not clearly show the difference. Please modify the figure or add a note in the caption about how the alignments of bilayers in these two cases are different.

- ✓ **Response (1)**

Thanks for this helpful comment. We have modified the caption to include the director alignments of bilayers.

“(c) Biomimetic bending ($\theta_u = 0^\circ$, $\theta_l = 90^\circ$) , spiral ($\theta_u = 120^\circ$, $\theta_l = 30^\circ$) and twisting shapes ($\theta_u = 135^\circ$, $\theta_l = 45^\circ$) can be flexibly achieved by designing the orientations of LCE director between both layers during DIW process.”

- **Comment (2)**

Below Eq. (7): “Figure 2(b) indicates that the twisting angle β nonlinearly varies with the director angle difference $\Delta\theta$, remarkably consistent with theoretical prediction in Eq. (7).”

The relation of β and $\Delta\theta$ is not clearly given in Eq. (7). Please clarify this.

- ✓ **Response (2)**

We thank the reviewer for this kind suggestion. We have added the definition of β in the main text, which gives the explicit relation between the twisting angle and director angle difference.

“The total twisting angle along the strip length is defined as $\beta = 180kL/\pi = 270(\alpha_\perp - \alpha_\parallel)\Delta TL \sin\Delta\theta/\pi H$.”

- **Comment (3)**

Fig. 2(c-d): Please add a clear note as to why the simulation and experiments deviate from theory at larger temperatures. Is there any relation between the deviation point and the nematic alignments in LCE layers? I recommend more analysis and comments on the physics of these observations.

- ✓ **Response (3)**

Thanks for this helpful suggestion. In the numerical simulation, we employed the same material constitutive law as in the theoretical analysis but considered 3D Green-Lagrange strains and full geometric nonlinearity, which yielded consistent tendencies with experiments. Therefore, we believe that the deviation at higher temperature (larger deformation) is primarily attributed to the simplified geometric configuration of the strip considered in the theoretical analysis.

We have added relevant discussions in the main text.

“Note that the difference between theory and experimental results at larger ΔT (Fig. 2(c)-(d)) can be attributed to the increasing geometric nonlinearity. As temperature arises, additional bending deformation along the width direction partially relieves the twisting or spiral deformation predicted by the theory.”

- **Comment (4)**

The results throughout the paper are mainly descriptive referring to the experimental figures. I recommend authors add more quantitative values to support their descriptions. For example “Once the twist stabilizes, rainfall was simulated. The raindrops absorb heat from the “leaves”, causing a decrease in leaf temperature and consequently inducing an untwisting deformation”: Please provide quantities on how much the twist and untwist are. And what are the actuation and relaxation times?

- ✓ **Response (4)**

Thanks for this kind suggestion. We have made the corresponding modifications by adding more quantitative values in the main text.

We have added the actuation and relaxation times for each stage in Fig. 4 and Fig. 5.

In Fig. 4:

In Fig. 5:

We have added the twisting angle and temperature changes to quantitatively describe the deformation process.

“As demonstrated in Fig. 4(a), the plant cluster initially exhibits a relatively flat leaf structure at room temperature 28.1°C.”

“Upon heating to 101.4°C, the “leaves” actively morph into twisting shapes like *Peisonia helix*, reaching an average twisting angle of 519.2° (Fig. 4(b)).”

“The raindrops absorb heat from the “leaves”, causing a decrease in leaf temperature to 86.9°C and consequently inducing an untwisting deformation to an average twisting angle of 274.6° (Fig. 4(c)).”

“Finally, we removed the heat source and allowed the sample to gradually cool down to room temperature, leading to further untwisting of the “leaves” back to the initial configuration with an average residual twisting angle of 26.2° (Fig. 4(d)).”

“The “leaf” under laser illumination was twisted to 210.9° because of thermal effect in the irradiation area, while the others remained unchanged.”

“As shown in Fig. 5(f), the cluster with untwisted “leaves” collects 2.4 ml of water under raining at room temperature 25.8°C . Upon heating to 79.0°C , leading to a spontaneous twisting of “leaves”, the amount of water collection increases to 7.3 ml over the same period.”

We have provided the duration of each water collection stage of Fig. 5(f)-(g) in Experimental setup.

“For adaptive droplet collection experiments in Fig. 5(f)-(g), each collection duration lasted 1 min.”

- **Comment (5)**

Also, “Finally, we removed the heat source and allow the sample to gradually cool down to room temperature, leading to further untwisting of the “leaves” back to the initial configuration”:

what is the time scale of twisting subject to illumination and untwisting after removing illumination?
What is the light intensity and wavelength?

- ✓ **Response (5)**

We have added the time scale information in Fig. 4, see Response (4). The light wavelength is 450 nm and was provided in the manuscript. The light intensities used in experiments have been added in the main text.

“We further demonstrated that such active twisting response can be flexibly tuned by localized laser (450 nm, 3.07 W/cm^2) illumination (Fig. 4(e)-(h)).”

“We further show in Fig. 5(g) the flexible tunability of morphing-affected droplet collection by localized laser illumination (1.65 W/cm^2 , the upper part of the cluster).”

- **Comment (6)**

Please also provide information about the phase transition temperature of the synthesized LCEs.

- ✓ **Response (6)**

Thanks for this kind suggestion. We have provided the phase transition temperature of LCEs in Section III of Supplementary Material.

“Note that the phase transition temperature of the LCEs is 105°C , which is consistent with the similar synthesis methods [5].”

[5] Ambulo, C.P., Burroughs, J.J., Boothby, J.M., Kim, H., Shankar, M.R., Ware, T.H., 2017. Four-dimensional printing of liquid crystal elastomers. *ACS Appl. Mater. Interfaces* 9, 37332–37339.

- **Comment (7)**

Please add the time scales in Fig. 4(a-h).

- ✓ **Response (7)**

We have added the time scales in Fig. 4(a-h), see Response (4).

- **Comment (8)**

The authors should make comments on why the initial twisted configuration of leaves in Fig. 5d and 5f are different. If it depends on temperature, the authors should mention the temperature.

- ✓ **Response (8)**

Thanks for this kind suggestion. The initial twisted configuration in Fig. 5(d) was specifically designed with a permanent twist configuration to enable direct comparison of wind resistance and water collection efficiency between twisted and flat shapes, which is temperature-independent. To clarify this, we have modified the corresponding expression in the main text. Besides, we have added the temperature changes for Fig. 5(f), see Response (4).

“We further looked into the collective effect on water collection by examining biomimetic permanently preset twisting clusters and bending shaped clusters. The comparisons between twisting and bending clusters in single-level (Fig. 5(c)) and multi-level (Fig. 5(d)) configurations indicate that the twisting clusters display a superior water-collecting performance (Supplementary Video S2).”

- **Comment (9)**

The authors should identify the temperatures or temperature changes in all experiments, particularly, in Fig. 4 and 5 (and any other figures as needed).

- ✓ **Response (9)**

We thank the reviewer for this kind suggestion. We have added the temperature changes for each stage for Figs. 4 and 5 in the main text, see Response (4).

- **Comment (10)**

The authors should identify the light intensity and wavelength used in experiments in all Figures.

✓ **Response (10)**

Thanks for this suggestion. The light intensity and wavelength have been added in all figures in the main text, see Response (5).

• **Comment (11)**

Please provide more details on mesh convergence in the Method section. What is the size of the converged mesh used in the simulation?

✓ **Response (11)**

Thanks for this kind suggestion. We have provided the converged mesh details in the Method section in the main text.

“The mesh convergence was carefully examined, and mesh density was set as $50 \times 4 \times 2$ for bending mode, $50 \times 8 \times 2$ for twisting mode, and $150 \times 10 \times 2$ for spiral mode, respectively.”

• **Comment (12)**

The authors should carefully review the equations, figures, and citations in SI for any potential typos. For instance, in SI, part C: “The general spatial orientation of LC director is denoted by a unit vector $\mathbf{n} = (\sin\phi\cos\theta, \sin\phi\sin\theta, \cos\phi)^T$, as shown in Fig. S1(b), where $\phi_l(u)$ denotes the angle between director and x_3 -axis, $\theta_l(u)$ is the angle between the projection of director on x_1Ox_2 plane and x_1 -axis.” Fig. S1(b) shows something different. Please also identify angles ϕ and θ in the figure for the lower layer.

✓ **Response (12)**

Thanks for this kind suggestion. We have carefully reviewed the equations, figures, and citations in Supplementary Material, and made the corresponding corrections.

“The general spatial orientation of LC director is denoted by a unit vector $\mathbf{n} = (\sin\varphi\cos\theta, \sin\varphi\sin\theta, \cos\varphi)^T$, as shown in Fig. S1(b), where $\varphi_l(u)$ denotes the angle between director and x_3 -axis...”

“Here, we consider the LC director aligned in the x_1Ox_2 plane (planar alignment), *i.e.*, $\varphi_l(u) = 90^\circ$, and thus the director vector is reduced to $\mathbf{n} = (\cos\theta, \sin\theta, 0)^T$, as shown in Fig. S1(e).”